# Glucose Stimulates Gut Motility in Fasted and Fed Conditions: Potential Involvement of a Nitric Oxide Pathway

**DOI:** 10.3390/nu14102176

**Published:** 2022-05-23

**Authors:** Eve Wemelle, Lionel Carneiro, Anne Abot, Jean Lesage, Patrice D. Cani, Claude Knauf

**Affiliations:** 1INSERM U1220, Institut de Recherche en Santé Digestive (IRSD), Université Paul Sabatier, Toulouse III, CHU Purpan, Place du Docteur Baylac, CS 60039, CEDEX 3, 31024 Toulouse, France; eve.wemelle@inserm.fr (E.W.); lionel.carneiro@inserm.fr (L.C.); 2NeuroMicrobiota, International Research Program (IRP) INSERM/UCLouvain, 31024 Toulouse, France; 3Enterosys SAS, 31670 Labège, France; anne.abot@enterosys.com; 4Université de Lille, Inserm, CHU Lille, U1286-INFINITE-Institute for Translational Research in Inflammation, F-59000 Lille, France; jean.lesage@univ-lille1.fr; 5UCLouvain, Université Catholique de Louvain, Louvain Drug Research Institute, WELBIO, Walloon Excellence in Life Sciences and BIOtechnology, Metabolism and Nutrition Research Group, 1200 Brussels, Belgium

**Keywords:** gut motility, glucose, enterosynes, NO, glucose transporters, type 2 diabetes

## Abstract

(1) Background: Type 2 diabetes (T2D) is associated with a duodenal hypermotility in postprandial conditions that favors hyperglycemia and insulin resistance via the gut-brain axis. Enterosynes, molecules produced within the gut with effects on the enteric nervous system, have been recently discovered and pointed to as potential key modulators of the glycemia. Indeed, targeting the enteric nervous system that controls gut motility is now considered as an innovative therapeutic way in T2D to limit intestinal glucose absorption and restore the gut-brain axis to improve insulin sensitivity. So far, little is known about the role of glucose on duodenal contraction in fasted and fed states in normal and diabetic conditions. The aim of the present study was thus to investigate these effects in adult mice. (2) Methods: Gene-expression level of glucose transporters (SGLT-1 and GLUT2) were quantified in the duodenum and jejunum of normal and diabetic mice fed with an HFD. The effect of glucose at different concentrations on duodenal and jejunal motility was studied ex vivo using an isotonic sensor in fasted and fed conditions in both normal chow and HFD mice. (3) Results: Both SGLT1 and GLUT2 expressions were increased in the duodenum (47 and 300%, respectively) and jejunum (75% for GLUT2) of T2D mice. We observed that glucose stimulates intestinal motility in fasted (200%) and fed (400%) control mice via GLUT2 by decreasing enteric nitric oxide release (by 600%), a neurotransmitter that inhibits gut contractions. This effect was not observed in diabetic mice, suggesting that glucose sensing and mechanosensing are altered during T2D. (4) Conclusions: Glucose acts as an enterosyne to control intestinal motility and glucose absorption through the enteric nervous system. Our data demonstrate that GLUT2 and a reduction of NO production could both be involved in this stimulatory contracting effect.

## 1. Introduction

The number of new patients suffering from type 2 diabetes (T2D) is still rising despite decades of extensive research on the causes of the disease. Among the most recent progresses, several recent advances have highlighted the role of the gut on this metabolic disorder [1]. In fact, T2D is associated with numerous intestinal dysfunctions strongly altering the gut-brain communication. Dysfunctions of the gut-brain axis communication were proposed to be at least in part involved in T2D onset. Indeed, several dysfunctions have been reported under T2D condition. Among them, intestinal glucose sensing is altered [2,3,4], leading to the genesis of an aberrant afferent nervous message that participates in the dysfunction of hypothalamic neurotransmitters release such as nitric oxide (NO) that is well characterized for its role in the control of glucose homeostasis [2,5]. In addition, diabetic mice present an alteration of the enteric nervous system (ENS) that provokes a duodenal hyper-motility leading to an increase in intestinal glucose absorption [6,7]. Similarly, diabetic humans also present ENS alterations such as a significant increase of *choline acetyltransferase* (*ChAT*) mRNA expression as well as a decrease of *neuronal nitric oxide synthase (nNOS)* mRNA expression in the duodenum [8]. These two enzymes are involved in the synthesis of acetylcholine and NO, which represent, respectively, the main stimulatory and inhibitory neurotransmitter of the ENS controlling intestinal smooth muscle cells’ contraction. We recently demonstrated that in T2D mice, such a similar alteration of the ENS leads to a duodenal hyper-motility. Furthermore, we also discovered that mechanosensors of the gut can sense this hypercontractility and trigger an inhibition of the release of hypothalamic NO followed by tissue glucose uptake to counter the hypercontractility [8,9,10]. Based on these observations, we have proposed to use bioactive molecules called ‘enterosynes’ to target the ENS activity in order to restore gut motility and gut-brain axis communication to treat T2D. An enterosyne from the gut could be defined as a molecule secreted by the host (enterocytes, enteric neurons, immune cells) or by microbiota that has an action on the ENS to control intestinal contraction [11]. For instance, our group has reported that molecules produced by enterocytes such as apelin, by the ENS such as galanin, or some products of the gut microbiota activity (enkephalins and bioactive lipids), can act as enterosynes to control glucose homeostasis and insulin sensitivity in diabetic mice [9,10,11,12].

Other molecules found in the gut environment could also represent yet unidentified enterosynes. For instance, some nutrients absorbed in the intestine could be considered as enterosynes since some of them are able to modulate both gut contractions and ENS activity. Indeed, it is well established that nutrients sensed at the level of the intestine trigger adaptive responses (cholecystokinin, gastrin, glucagon-like peptide 1) that play a role on gut contractions and metabolic regulation [9]. In fed conditions, gut contractions participate in the glucose absorption rate by increasing the contact between this nutrient and the enterocyte of the intestinal wall [13]. However, so far, the capacity of glucose to control the ENS and intestinal contractions rhythmicity in the upper part of the small intestine where glucose is mainly absorbed has never been investigated. Here, we aimed to investigate the interplay between glucose detection and gut contraction, from a fasted to fed state, and address whether glucose could be considered as an enterosyne. We hypothesize that alteration of intestinal glucose detection in the “diabetic gut” could be associated with alteration of gut motility that is known to be correlated with an insulin resistance state. Targeting the glucose signaling pathway could be an innovative therapeutic approach since it was recently discovered as having an impact of a dual blockade of sodium-glucose co-transporter-1 and 2 (SGLT1/2) on post-prandial glycemia by delaying glucose absorption [14].

Consequently, we investigated the impact of glucose on gut contraction in physiological and pathological conditions, i.e., diabetic mice fed with a high-fat diet. In this study, we aimed to decipher the mode of action of glucose by measuring the effect of glucose on enteric NO release and the impact of the blockade of intestinal glucose transport on gut motility.

## 2. Materials and Methods

### 2.1. Mice

For in vivo experiments, nine-week-old male C57BL/6J mice (Charles River Laboratory, L’Arbresle, France) were used. Mice were housed in a specific pathogen-free, light- and temperature-controlled environment (room temperature of 23 ± 2 °C, 12 h daylight cycle). All animals were given free access to food (normal chow (NC) diet or high-fat diet (HFD) containing 20% protein, 35% carbohydrate, and 45% fat (Research Diet, New Brunswick, NJ, USA)) and water. Mice fed a 45% high-fat diet presented an obese/diabetic phenotype including an increase in body weight, fasted hyperglycemia and insulinemia, glucose intolerance, and insulin resistance as previously published [8,10,15]. After a one-week acclimatization period, experiments were conducted according to the European Community regulations concerning the protection of experimental animals and were approved by the local Animal Care and Use Committee under the protocol number 2021042609281581.

### 2.2. Isotonic Contraction

For gut contraction analysis, mice were euthanized in fed conditions or after a 10 h fasting at the end of a 3-month period of feeding with NC or HFD (Figure 1A). Amounts of 1.5 cm of duodenum or jejunum were quickly dissected and washed in Krebs–Ringer bicarbonate/glucose buffer (pH 7.4) in 95% O_2_ and 5% CO_2_. All the segments were then incubated in oxygenated Krebs–Ringer solution for 30 min at 37 °C. After the incubation, each segment was attached to an isotonic transducer (MLT7006 Isotonic Transducer, Hugo Basile, Comerio, Italy), and immersed in tubes containing 25 mL of Krebs–Ringer solution at 37 °C. The load applied to the lever was 2 g (20 mN). Isotonic contractions were recorded on Labchart software (AD Instruments, Sydney, Australia) following the transducer displacement. After attaching the gut segments, basal contractions were recorded for 10 min. Subsequently, 100 μL of Krebs–Ringer solution or specific drugs (glucose concentrations of 1, 2, 5.5, 10 to 20 mM and phloretin 0.5 mM after glucose 20 mM priming) were added to the survival medium, and contractions were recorded for another 10 min. Amplitudes were recorded every 10 s during 10 min and the average was compared to the average basal contractions. Contraction amplitudes and frequencies are presented as percentage relative to the basal response and contraction frequencies are presented as number of contractions per minute as previously described [8,10,15].

### 2.3. Gene Expression

After the 3-month NC or HFD period, fed mice were euthanized, and gut segments were quickly harvested and snap frozen in liquid nitrogen for further processing. For gene expression, tissues were homogenized using a Precellys tissue homogenizer (Bertin technol., Montigny-le Bretonneux, France). Total RNA was extracted using TRIReagent (Sigma-Aldrich, L’lsle-d’Abeau Chesnes, France) and GenElute mammalian Total RNA Miniprep Kit (Sigma-Aldrich), according to the manufacturers’ instructions. cDNA was generated using a Moloney murine leukemia virus reverse transcriptase (M-MLV), a reverse transcriptase kit (Invitrogen, Waltham, MA, USA), and random hexamers (Invitrogen). The real-time quantitative PCR was performed using SYBR Green Real Time PCR Master Mixes (Thermo Fisher Scientific, Waltham, MA, USA). Amplifications and quantifications were done in a LightCycler 480 (Life Technologies, Carlsbad, CA, USA). Primers used for cDNA amplification in the quantitative PCR experiments were first tested for PCR efficiency and are listed in Table 1. Gene expression was quantified using the ^∆∆^CT method with HPRT1 as a housekeeping control gene. Finally, the identity and purity of the amplified product were assessed by analyzing the melting curve.

### 2.4. Longitudinal Muscle and Adherent Myenteric Plexus (LMMP) Dissection and Myenteric Neuronal Cells Culture

Nine-week-old mice were euthanized, and the abdomen was longitudinally opened to expose the small intestine. The proximal part of the intestine (from the duodenum to the ileum) was cut, rinsed with HBSS, and placed in LMMP dissection medium (sterile HBSS Ca2^+^ (Sigma-Aldrich) + Penicillin/streptomycin (100 µg/mL) (Sigma-Aldrich) + Gentamycin (50 µg/mL) (Sigma-Aldrich)) on ice. Once dissected, each section was then cleaned from remaining mesentery. The intestinal section was then inverted on a short stick and the LMMP was detached using a sterile cotton-tipped applicator wet in cold LMMP dissection medium. The LMMP was then stored in LMMP dissection medium and dissected in small pieces for further processing [16].

Once dissected, the LMMP were collected and transferred to a 15 mL tube and centrifuged for 5 min at 200 g and room temperature. The supernatant was discarded and 2 mL/mice of LMMP digestion medium (LMMP dissection medium + collagenase (1 mg/mL) (Serlabo, Entraigues-sur-la-Sorgue, France) + 10 mM HEPES (10 mM) (Sigma-Aldrich)) at 37 °C were added. LMMPs were then incubated at 37 °C for 25 min with a gentle resuspension every 5 min. After the incubation, LMMP solution was centrifuged for 5 min at 200 g and RT and the supernatant was removed. LMMPs were then washed twice using 10 mL of LMMP dissection medium to resuspend the LMMPs followed by a centrifugation for 2 min at 200 g and RT. The supernatant was removed and 1 mL of Trypsin (Sigma) was added and was incubated at 37 °C for 7 min with gentle mixing every 2 min. Then, a mechanical dissociation by pipetting up and down was performed for 3 min and 7.5 mL of complete DMEM (Sterile DMEM F12 (Sigma-Aldrich) + 10% SVF (Sigma-Aldrich) + Penicillin/streptomycin (100 µg/mL) + Gentamycin (50 µg/mL)) were added. After a centrifugation for 5 min at 200 g and at RT, the pellet was resuspended in 400 µL of non-mitotic neurobasal medium (sterile Neurobasal A (Thermo Fisher Scientific) + B27 (Thermo Fisher Scientific) + L-glutamine (200 mM) (Thermofisher) + Penicillin/streptomycin (100 µg/mL) + Gentamycin (50 µg/mL) + 5-Fluoro-2-Deoxyuridine (50 mM) (Sigma-Aldrich) + Uridine (50 mM) (Sigma-Aldrich)) and plated on a Labtek II chamber. Non-mitotic neurobasal medium was replaced every 2 days for 7 days before NO measurement to eliminate non-neuronal cells.

### 2.5. NO Real Time Measurement in Neuronal Cells

Neuronal cell cultures were removed, and cells were washed twice with Krebs–Ringer bicarbonate/glucose buffer (pH 7.4) in 95% O_2_ and 5% CO_2_. Before the experiment, the medium was changed and after a 10 min recovery period, the spontaneous NO release was measured at 37 °C for 5 min in culture wells by using a specific amperometric probe (ISO-NOPF, 100 μm diameter, 5 mm length, World Precision Instruments, Aston Stevenage, UK) in response to glucose concentrations of 5.5 mM. For NO measurement, 2 mice were needed per well to have one point. A total of 22 mice (10 for control and 11 for glucose-treated cells) were used for this experiment.

### 2.6. Statistics

Results were analyzed using the GraphPad Prism 9.3.0 software (San Diego, CA, USA). Outliers were detected via a Grubb’s test. Normality of variables was then assessed using a D’Agostino Pearson test. Finally, statistical differences were tested by paired *t*-test or one-way ANOVA; the test was according to sample normality assessment results. Variations were considered as significant when *p* value was <0.05. Results in the article are presented as mean ± SEM.

## 3. Results

### 3.1. HFD Increases the Expression of Glucose Transporters in the Intestine

HFD induced a significant increased gene-expression level for GLUT2 (*SLC2a2*) and SGLT1 (*SLC5a1*) in the duodenum (Figure 1B). Although, the jejunum part only displays an increase of *SLC2a2* mRNAs (Figure 1C).

### 3.2. Under Normal Diet Conditions, Glucose Induces Changes in Duodenal and Jejunal Contractility via a GLUT2-Dependent Mechanism

In fasted state, the duodenum contractions are stimulated by glucose concentration starting at 5.5 mM. A lower glucose concentration has no effect while higher concentrations do not further increase the contractility. Both frequency and amplitude are stimulated by glucose in duodenum (Figure 2A). However, no stimulatory role of glucose was observed for jejunum contractions (Figure 2B).

In fed conditions, both duodenum and jejunum showed an increase in contraction amplitude after glucose stimulation, while frequencies were unaffected. Furthermore, the stimulatory effect also starts for a 5.5 mM concentration of glucose, without any additional effect of a higher dose (Figure 2C,D).

To investigate if glucose transporters are implicated in the stimulatory effect of glucose on gut contractions, we added in our ex vivo model phloretin, a pharmacologic GLUT2 inhibitor. Using this strategy, we observed that the inhibition of GLUT2 by the addition of phloretin after glucose abolished the stimulatory effect of glucose on both duodenal or jejunal contractions (Figure 2E–H).

### 3.3. Glucose Decreases NO Release from Neuronal Cells of the LMMP In Vitro

Enteric NO is a main inhibitor neurotransmitter of intestinal contractions whose concentrations are strongly altered in the gut during T2D. Therefore, we wanted to measure the NO release in response to glucose in vitro using neurons isolated from mouse small intestines. Results show that a 5.5 mM glucose addition induces a drastic decrease of NO released from neuronal cells (Figure 3).

### 3.4. HFD Is Associated with a Loss of Glucose-Induced Contractility

We show that neither duodenal nor jejunal sections display any change in contractility following a glucose stimulation in HFD mice (Figure 4A,B). However, when GLUT2 was inhibited using phloretin, both duodenum and jejunum were impacted by the treatment. Indeed, duodenum contraction amplitude was decreased, while in jejunum, both amplitude and frequency were decreased compared to the controls (Figure 4C,D).

## 4. Discussion

Among the different molecules that can be classified as enterosynes, hormones, bioactive peptides, or lipids, nutrients, microbiota, and immune factors have been identified [9]. These enterosynes are of interest in metabolic control since T2D is characterized by an hypermotility measured in the intestine. These increased gut contractions favor glucose absorption, which in turn contributes to hyperglycemia [10,17]. Finally, the regulation of the ENS to restore a normal motility leads to a recovery of a normal gut-brain axis communication in addition to an improved insulin sensitivity [9]. Altogether, targeting gut contractions is a promising strategy for diabetic subjects in order to modulate glucose homeostasis and control hyperglycemia [18,19,20].

Here, we first wanted to assess the role of glucose on gut motility in fasted conditions (i.e., at the beginning of a meal) as well as in fed conditions.

Interestingly, in fasted conditions, the addition of glucose stimulates duodenal contractions by increasing both amplitude and frequency. Such a result suggests that glucose acts as a synchronizer of contractions in a pre-prandial situation. It is most likely that glucose induces the synchronization of enteric neurons activity with smooth muscles and interstitial cells of Cajal (ICC) already known to contribute to gut contractions [21,22,23]. Such a putative role is supported by the threshold of contractions stimulation observed at 5.5 mM without an additional effect for higher concentrations of glucose. In such conditions, glucose could act to stimulate contractions in order to facilitate absorption of nutrients present in a meal. This observation is in accordance with the work of Sababi and Bengtsson showing a positive correlation between gut motility and glucose absorption [13]. However, the precise mechanism leading to this coordination has yet to be elucidated. Nevertheless, NO production is decreased by enteric neurons, suggesting a direct sensing of glucose by nNOS neurons. Glucose is known to target enteric nNOS neurons from the duodenum and the jejunum [24], but we are the first demonstrating the impact on NO release. In addition, NO is able to decrease gut contractions by acting on both smooth muscle cells and ICC [25]. Finally, NO measurements showed that effects of glucose on duodenal and jejunal contractions depend on the inhibition of NO release from neuronal cells. It is not surprising to have such an effect since enteric NO is the main target of previously described enterosynes.

To go further, we tested the effect of glucose on gut motility in fed mice. In such conditions, glucose is still able to increase gut contractions by stimulating the amplitude on duodenum and jejunum. In addition, we also observed such a stimulatory effect on both duodenum and jejunum sections. Altogether, this result suggests that in a fed state, glucose stimulates gut motility in order to increase the intestinal glucose absorption during a meal.

Accordingly, T2D, a condition associated with gut hypermotility, also shows an increased level of GLUT2 expression at the apical side of the enterocytes as well as an increased SGLT1 expression. This increased expression is due to an increase in the translocation of the transporter [26,27,28,29]. Interestingly, we noticed an increase in both transporter expression in the duodenum, while the jejunum only shows a GLUT2 overexpression supporting a role in the duodenal phenotype observed in T2D [30]. Overall, this increase would lead to a greater glucose uptake by the enterocytes, and thus, an increase of glucose absorption. Interestingly, the inhibition of glucose transporters using phloretin completely blocks the effects of glucose in both duodenum and jejunum. Phloretin has the highest affinity for GLUT2, thus supporting a role for this transporter in the motility changes induced by the glucose.

The last aim of our study was to test the effect of glucose on gut contractions in a model of diabetes induced by a HFD 45% [2]. In such conditions, glucose has no impact on gut motility. This result is consistent with a loss of glucose sensing as described in previous studies [2,4]. Actually, dysregulations of the gut-brain axis are strongly associated with T2D [1]. In this key regulatory loop, glucose sensing in the intestine plays a central role by controlling the gut-brain axis. Consequently, a dysregulation in gut glucose sensing will also disturb the gut-brain axis. This enteric glucose sensing appears to be at the center of the glucose homeostasis since it triggers different signals that will impact blood glucose and energy homeostasis. Furthermore, other reports demonstrate that an alteration of the intestinal glucose detection disturbs the brain control of energy homeostasis [2,9,17,31].

In our model of HFD-45%-induced diabetes, the expression analysis of glucose transporters in the duodenum and jejunum revealed an increased expression of GLUT2 in duodenum and jejunum, while SGLT1 increases only in the duodenum. This result suggests an alteration of glucose uptake during the HFD, that could participate in the glucose sensing defect described in HFD-induced obesity. Overall, this finding suggests that glucose sensing modification takes its origin within the gut. In addition, no motility changes were measured after the addition of glucose in the intestinal sections from HFD mice. Nonetheless, phloretin treatment reduces duodenal and jejunal contractions, thereby indicating that GLUT2 may play a role in gut contraction during the HFD. We may not rule out that the gut contraction is at full capacity after an HFD-induced obesity as we observed previously [8,10,15]. Thus, the addition of higher doses of glucose would have no additional effect, confirming the alteration of glucose sensing in the diabetic gut [2]. Here, our results support a role of gut motility on the uncontrolled post-prandial hyperglycemia observed during diabetes. In fact, the hyperactivity of the intestine could favor a greater uptake of glucose and alter the gut-brain axis to favor insulin resistance. Management of post-prandial hyperglycemia of diabetic patients by acting on glucose transporters such as SGLT-1 [14] or GLUT2 in this present paper could represent a real opportunity, first to decrease glucose absorption and second to restore the gut-brain axis to treat insulin resistance. Future experiments will focus on this novel strategy.

## 5. Conclusions

Our results provide evidence that diabetes could be characterized by defects in both chemosensing and mechanosensing of glucose within the gut. In the present study, we show that higher glucose concentration plays a role in the gut motility in both fasted and fed states.

Therefore, new strategies aiming at improving the glucose sensing and signaling in the gut could be of interest in future treatment of diabetic patients. Targeting GLUT2 to modulate gut motility in diabetes in fed conditions could be a future therapeutic strategy to consider.

## Figures and Tables

**Figure 1 nutrients-14-02176-f001:**
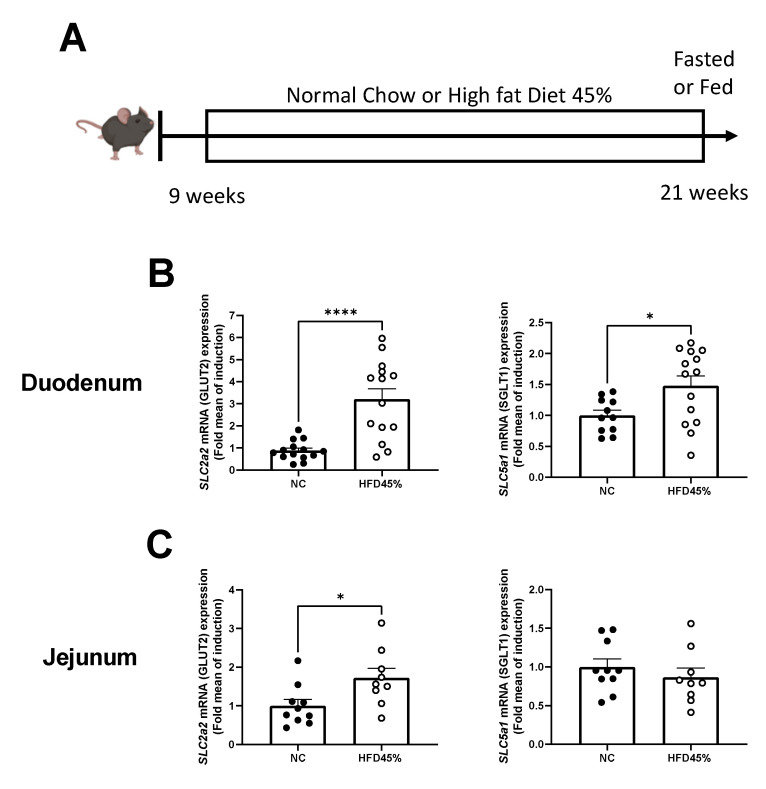
HFD 45% increase *SLC2a2* (GLUT2) and *SLC5a1* (SGLT1) mRNA levels in mice duodenum and SLC2a2 in mice jejunum. (**A**) Schematic representation of in vivo experiments. mRNA expression of SLC2a2 and SLC5a1 in the duodenum (**B**) and jejunum (**C**) in control mice or diabetic mice fed a HFD 45%. *n* = 14 NC and *n* = 14 HFD 45%. * *p* < 0.05, **** *p* < 0.0001.

**Figure 2 nutrients-14-02176-f002:**
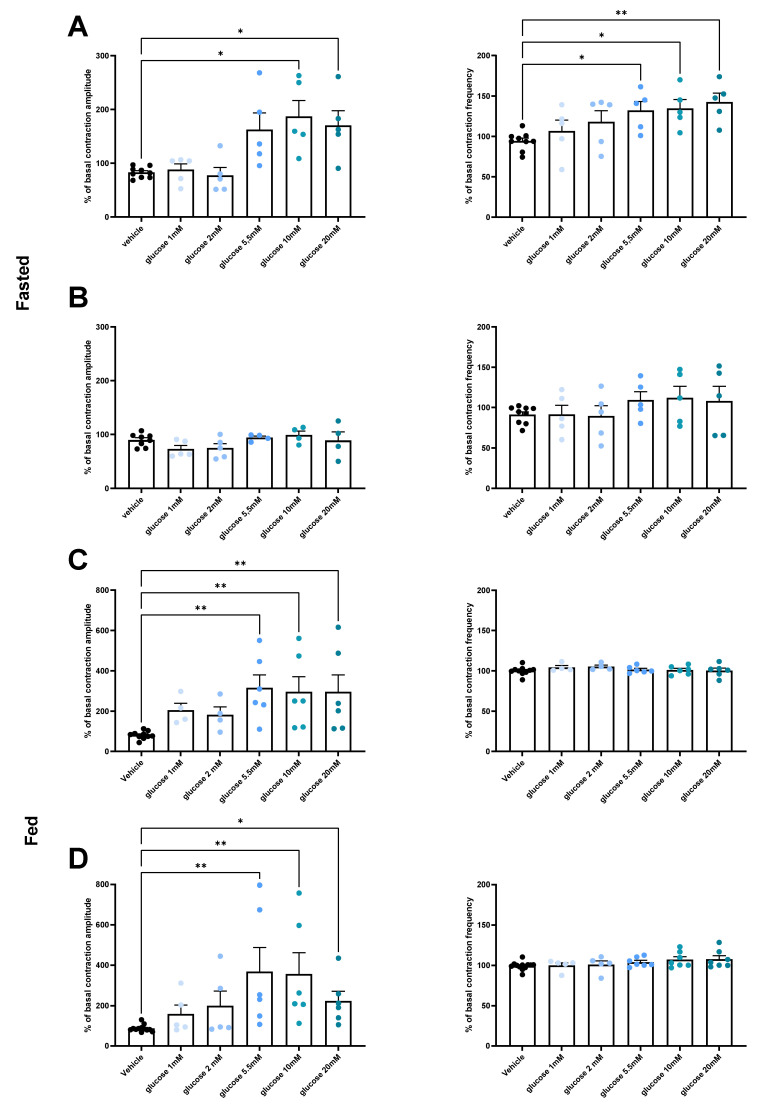
Glucose increases duodenal and jejunal contractility in control mice. Ex vivo measurement of the duodenal (**A**) (*n* = 9 vehicle, *n* = 5 for each glucose concentration) and jejunal (**B**) (*n* = 8 vehicle, *n* = 5 for glucose 1 and 2 mM, *n* = 4 for glucose 5.5, 10, and 20 mM) mechanical contraction amplitude and frequency in response to Krebs–Ringer (vehicle) or glucose (1, 2, 5.5, 10 or 20 mM) and phloretin (**E**) (*n* = 5 for each condition) and (**F**) (*n* = 5 for vehicle and Phloretin groups, and *n* = 4 for glucose 20 mM group) in fasting NC mice, and duodenal (**C**) (*n* = 10 vehicle and *n* = 4 glucose 1 and 2 mM, and *n* = 6 for glucose 5.5, 10, and 20 mM) and jejunal (**D**) (*n* = 12 vehicle and *n* = 5 glucose 1 and 2 mM, and *n* = 6 for glucose 5.5, 10, and 20 mM) mechanical contraction in fed mice in response to Krebs–Ringer (vehicle) or glucose (1, 2, 5.5, 10 or 20 mM) and phloretin (**G**) in duodenum and (**H**) in jejunum (*n* = 6 for each condition). * *p* < 0.05, ** *p* < 0.01, *** *p* < 0.001.

**Figure 3 nutrients-14-02176-f003:**
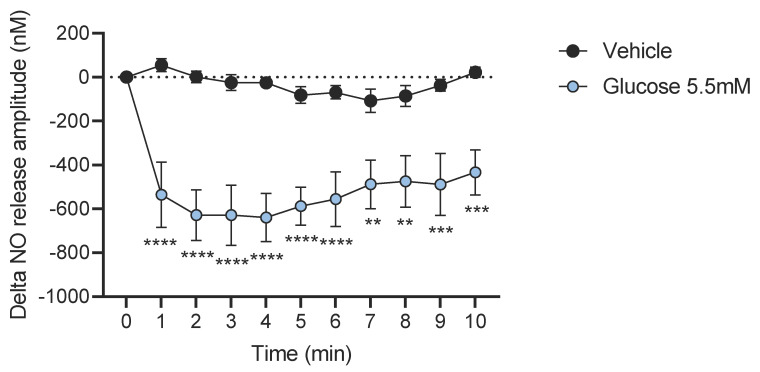
Glucose decreases NO release from myenteric neurons. Ex vivo measurement of NO release measured by real-time amperometric detection in primary neurons culture from myenteric plexus of control mice after injection of glucose (5.5 mM, *n* = 6) or Krebs–Ringer (vehicle, *n* = 5). ** *p* < 0.01, *** *p* < 0.005, **** *p* < 0.0001.

**Figure 4 nutrients-14-02176-f004:**
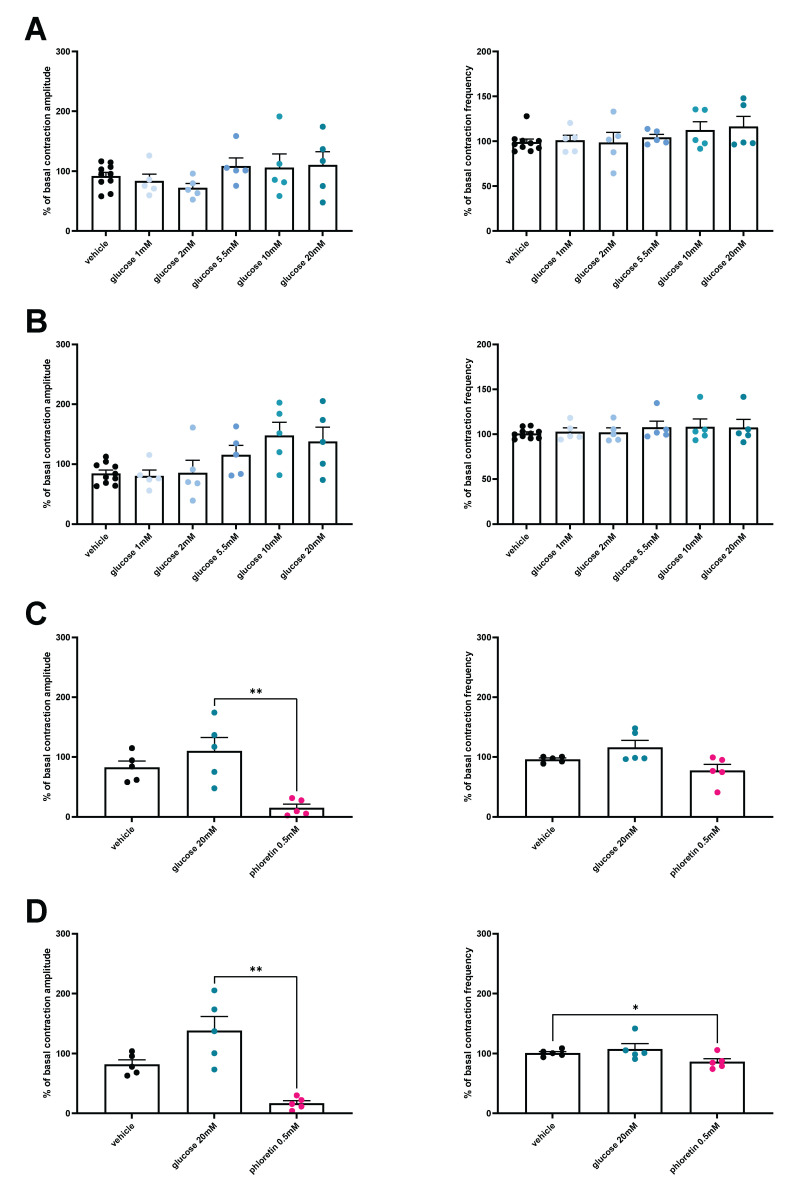
Loss of glucose effect on duodenal and jejunal contractility in HFD 45% diabetic mice. Ex vivo measurement of the duodenal (**A**) and jejunal (**B**) mechanical contraction amplitude and frequency in response to Krebs–Ringer (vehicle) (*n* = 10) or glucose (1, 2, 5.5, 10, or 20 mM, *n* = 5 for each concentration) and phloretin (*n* = 5 for each condition) (**C**,**D**) in fed HFD 45% mice. * *p* < 0.05, ** *p* < 0.01.

**Table 1 nutrients-14-02176-t001:** Primer’s list for glucose transporters gene expression quantification.

Gene	Forward Primer	Reverse Primer
*Hprt1*	gccagactttgttggatttgaa	gcttgcgaccttgaccatct
*Slc2a2 (GLUT2)*	tggaaggatcaaagcaatgttg	catcaagagggctccagtcaa
*Slc5a1(SGLT1)*	ggcttctccacctgcctcata	tagggcatccaggagatggtgt

## Data Availability

Data of this study are available upon request to authors.

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
