# Peer review of "Glucose Stimulates Gut Motility in Fasted and Fed Conditions: Potential Involvement of a Nitric Oxide Pathway"

_nutrients, 2022, doi:10.3390/nu14102176_

Round 1

Reviewer 1 Report

Title: Glucose stimulates gut motility via an inhibition of enteric nitric oxide release.

Dear Authors,

Congratulations on conducting this interesting study. This manuscript aims to reveal the mode of action of glucose by measuring the effect of glucose on enteric NO release and the impact of the blockade of intestinal glucose transport on gut motility. The gut-brain axis dysfunction is well known, and the author successfully has reached the conclusion of their hypothesis. 

The following are my comments and suggestions:

Abstract and Introduction:

  1. My most of comments are on the presentation and consistency of data presentation. The manuscript needs to be edited heavily for data presentation. Kindly provide the results about diabetic condition establishment in your rodent model.
  2. Abstract. This is too wordy and unhelpful to conclude changes among parameters. Please provide changes of parameters in percentage form to show the degree of changes that appeared in the data.
  3. The introduction section needs to expand so that reader can illustrate the information in the introduction what as the problem, history, and Author's hypothesis.   

Methods:
1. What was the acclimatization period for mice before starting of protocol? Did the author check bodyweight on regular basis? Why did bodyweight data not presented as HFD diet was involved there? Did diabetic condition establish in the mice model? OGTT, insulin test, AUC, and biochemical tests were conducted to prove the diabetic model?

With the help of histology and IHC, the author should show the relevance of the intestine sample study that was used in this project.

2. Please mention the total number of mice used in these experiments and their distribution into the different groups before termination.

3. 2.2 – Please mention the size of tissue used in the isotonic contraction experiment. 

4. 2.4 – Please provide the reference for this method.

Results:
1. Fig 2 – please increase the font size same as fig 1.
2. 3.3 – please mention what part of the small intestine was used here. Ileum? What about other parts of the intestine? Did they show similar results as well?

Discussion:
So, based on these conclusions, how do we proceed with managing hyperglycemia in the diabetic subjects? 

Please check the References. Years are not written the same way throughout the reference section 

Author Response

Our answers are in the pdf files.

Best regards,

Reviewer 2 Report

This manuscript tried to investigate the effect of high fat diet feeding on glucose induced intestinal contraction in ex vivo. It is interesting to focus ‘enterosynes’. However, I have some concern as follows in this manuscript. Some methods were not validated, experimental control was missing and critical results that draw the conclusion is not sufficient. It is unclear how the results obtained in this study link changes of physiological function in HFD fed mice.

  1. The title sounds overestimate.

There is no data that show gut motility was inhibited by inhibition of enteric NO release.

  1. Figure 2

Please show sample of raw data. I could not understand what were amplitude and frequency?

It is not clear how the contraction data presented in this manuscript link physiological function in the intestine.

What does 5.5 mM glucose-induced contraction physiologically mean?

What does phloretin experiment means? Did authors add phloretin with glucose or not? If with glucose, phloretin without glucose group was missing. If without glucose, how phloretin regulate contraction?

I think sample size was too small in Fig2 and 4. Is there any evidence that the sample size is appropriate?

  1. Figure 3

Please show sample of raw data. I could not understand how delta NO release amplitude was revealed?

Authors should show evidence that these cells are functional myenteric neurons.

Author Response

Our answers are in the pdf file.

Best regards,

Round 2

Reviewer 2 Report

My recommendation of this manuscript is “Major revision”.

I think this study have not include adequate methods and critical control in results to draw the conclusion.

Author Response

Thanks for the novel comments from reviewer #2.
Reviewer #2 has proposed novel experiments that are different from the
first revision (MTT assay), which is impossible to answer to reviewer'
comments in 10 days as asked in the first round of revision. One
experiment proposed by reviewer #2 concerning phloretin could be
justified, but not essential for our study.This article is a short
communication, with the advantage of scientific novelty (and impact)
while having the defects of this type of articles, namely incomplete. I
gave all the arguments possible to reviewer #2 to demonstrate the
importance of our study, in this format.